# Fast Salient Factor Concentration (FSFC) Recurrent Neural Network for Text Classification

## Abstract

Models based on Recurrent Neural Networks (RNNs) have been widely employed for text classification tasks. Traditional RNNs primarily emphasize long-term memory capabilities. However, this approach does not fully align with human cognitive learning processes, particularly in the context of classification tasks. The human brain typically extracts essential information relevant to the classification categories, disregards irrelevant details, and compresses the input to accelerate decision-making. Inspired by this, we propose a novel architecture, the Fast Salient Factor Concentration (FSFC) RNN, specifically designed for classification tasks. FSFC dynamically clusters and compresses semantic information by leveraging the short-term memory capabilities of recurrent neural networks. Experimental results demonstrate that FSFC achieves performance comparable to existing RNNs, while significantly improving training efficiency in classification tasks. Based on the YelpReviewFull dataset, FSFC improves accuracy by 1.37% over Long Short-Term Memory (LSTM), while reducing training time by 86%. Additionally, we propose a new evaluation metric, E-score, which integrates both accuracy and time efficiency to comprehensively assess the overall performance of each network.

## 1 Introduction

Text classification is an important and fundamental problem in the field of natural language processing (NLP) (Du et al., 2020; Joulin et al., 2016; Magalhães et al., 2023; Wang et al., 2023b), with wide applications such as spam filtering, sentiment analysis, and news categorization (Wang et al., 2018; Yao et al., 2019; Zeng et al., 2018). With the advancement of deep learning technologies, numerous deep learning models have been introduced into text classification tasks. Recurrent Neural Networks (RNNs), especially Long Short-Term Memory networks (LSTM) (Hochreiter & Schmidhuber, 1997) and Gated Recurrent Units (GRU) (Cho, 2014), have garnered significant attention in the field of text classification (Liu & Guo, 2019; Luan & Lin, 2019).

However, in classification tasks, the long-term memory mechanisms of traditional RNNs do not fully align with human cognitive learning processes. When processing long texts or audio, humans typically rely on short-term memory, focusing on task-relevant key information while ignoring irrelevant content. Through selective attention mechanisms, working memory prioritizes important information and dynamically adjusts the focus and granularity of information processing (Hu et al., 2024; Jeanneret et al., 2023). This ability to compress and organize information allows humans to make more efficient decisions in complex tasks (Hu et al., 2024).

Inspired by cognitive mechanisms, we propose a novel RNN architecture specifically designed for classification tasks, called Fast Salient Factor Concentration Recurrent Neural Network (FSFC). Unlike traditional RNNs that predominantly rely on long-term memory (Duarte & Berton, 2023; Lu et al., 2023; Soni et al., 2022), FSFC fully exploits the short-term memory capabilities of RNNs while simplifying the network by removing complex gating mechanisms, leading to a significant improvement in computational efficiency. Moreover, FSFC enhances the processing of crucial information by employing dynamic clustering and semantic compression techniques. Experimental results indicate that FSFC achieves performance on par with existing RNN models, while consid-

erably reducing training time in classification tasks. The primary contributions of this work are as follows:

1. We propose a novel RNN architecture, FSFC (Fast Salient Factor Concentration), developed as an alternative to traditional RNN components. By integrating a semantic segmentation and clustering mechanism, FSFC effectively compresses textual information while utilizing the short-term memory capabilities of RNNs, leading to a significant enhancement in the efficiency of classification tasks.

2. We introduce a cognitive function for FSFC, inspired by the human learning process that transitions from detailed analysis to simplification, allowing for dynamic adjustment of the granularity of semantic clustering.

3. We design the E-score metric, which integrates classification accuracy and train time, providing a comprehensive evaluation of model performance.

## 2 RELATED WORK

### 2.1 TRADITIONAL TEXT CLASSIFICATION

Traditional research in text classification primarily focuses on feature engineering and classification algorithms (Yao et al., 2019). In early studies, conventional machine learning methods, such as Support Vector Machines (SVM) (Zhang et al., 2010) and logistic regression (Genkin et al., 2007), relied on sparse representation techniques, including the Bag of Words (BoW) model and TF-IDF. These methods classify text by converting it into word frequency or weighted frequency matrices. However, sparse representations fail to capture the contextual relationships between words, leading to significant limitations when handling complex texts (Wang et al., 2024).

To address the limitations of sparse representations, (Mikolov, 2013) introduced the Word2Vec model, which utilizes a Skip-gram architecture to embed words into a high-dimensional vector space via neural networks, thereby capturing local contextual information within the text. Each word's embedding vector carries rich semantic information, and the cosine distance between these vectors can effectively measure semantic similarity. Building on the Word2Vec model, researchers have proposed various improved embedding models, such as GloVe (Pennington et al., 2014), Doc2Vec (Le & Mikolov, 2014), and fastText (Xiong et al., 2021). These models enhance the understanding of textual semantics through more complex structured representations.

Unlike traditional word embedding models, FSFC employs a dynamic adjustment approach in its embedding layer. Traditional models are static and typically require pre-constructed corpora, demonstrating poor adaptability to new texts. In contrast, FSFC embedding layer continuously adjusts embedding vectors during training based on the loss function, analogous to how the human brain refines its understanding of new information. Dynamic embedding method enhances the model's adaptability and reduces its reliance on pre-trained corpora.

### 2.2 SEQUENTIAL MODELS FOR TEXT CLASSIFICATION

Neural networks based on GRU and LSTM architectures are mainly applied to learn multiple rich-semantic sequential information in the relationships between words and their belonged documents (Pham et al., 2022; Liu et al., 2016). (Kumar & S, 2022) proposed a hybrid model that combines Convolutional Neural Networks (CNN) (LeCun et al., 1989) with Long Short-Term Memory networks (LSTM) to improve short text classification performance. CNNs extract spatial features from the text, while LSTMs handle temporal sequence features, effectively capturing both local information and sequential dependencies in short texts. (Du et al., 2020) introduced an efficient recurrent neural network architecture based on Broad Learning System (BLS) (Chen & Liu, 2017), known as R-BLS and G-BLS, which are similar to LSTM architectures. By incorporating BLS, this architecture significantly accelerates training speed and mitigates common issues such as gradient vanishing and explosion typically associated with RNNs and LSTMs. R-BLS addresses the limitations of traditional BLS in processing sequential information and word importance, while G-BLS further enhances information processing capabilities by introducing LSTM-like forget gates, enabling the network to retain relevant information while discarding irrelevant data. (Behzadidoost et al., 2024) proposed a stacked BILSTM-SVM model that integrates Bidirectional Long Short-Term Memory

networks (BILSTM) (Schuster & Paliwal, 1997) with Support Vector Machines (SVM). This model merges the two using a stacked approach to enhance text classification performance. The bidirectional LSTM captures contextual information from both forward and backward directions, extracting deep semantic features (Lu et al., 2023), while the SVM utilizes the high-dimensional semantic features extracted by BILSTM for final classification.

Although the impressive performance of LSTM and GRU based models in text classification tasks (Nithya et al., 2024), they still exhibit limitations when handling long texts. The gating mechanisms of LSTM and GRU are primarily designed to capture long-term dependencies (Lu & Xu, 2023; Fathnejat et al., 2023; Jiang et al., 2023). However, in text classification tasks, models often need to focus only on key information relevant to the categories rather than all details within the text. This reliance on long-term memory may result in models capturing a significant amount of irrelevant information during lengthy text processing, thus reducing training efficiency and increasing computational overhead. Furthermore, these architectures still struggle to completely mitigate the prevalent issues of gradient vanishing or explosion found in RNNs (Reusens et al., 2024). These challenges suggest that relying solely on long-term memory RNN architectures may not be entirely suitable for text classification tasks.

## 3 METHODOLOGY

To address the inefficiency caused by the processing of redundant information in traditional RNNs for text classification tasks, we propose the Fast Salient Factor Concentration Recurrent Neural Network (FSFC). This model focuses on short-term memory and is capable of dynamically aggregating and compressing semantic information from the text, thereby reducing the computational load and accelerating the classification process. FSFC is inspired by cognitive mechanisms in the human brain, where essential task-related features are swiftly extracted in complex informational environments, while less relevant details are disregarded (Fonollosa et al., 2015).

FSFC consists of four stages: text mapping, semantic segmentation, clustering and compression, and category classification. Our experiments demonstrate that FSFC not only achieves accuracy comparable to traditional RNNs(LSTM, GRU) but also significantly improves training efficiency in text classification tasks. Figure 1 illustrates the operational mechanism of FSFC.

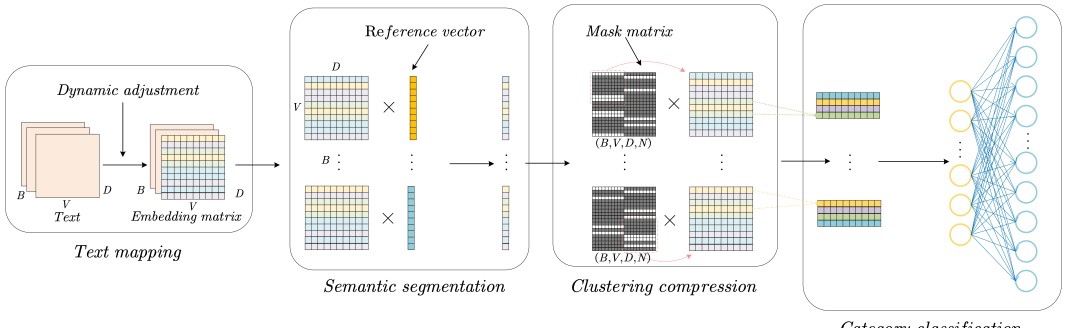

Figure 1: Operational Mechanism of FSFC.

### 3.1 TEXT MAPPING AND SEMANTIC SEGMENTATION

The input text first passes through the embedding layer, which randomly maps each word into a high-dimensional real-valued vector space (Shen et al., 2018; Defferrard et al., 2016). The vector space is dynamic, and the embedding layer adjusts the word embeddings based on the gradients of the loss function, thereby learning word representations that are better suited to the current task. Essentially, the embedding layer is a weight matrix, where each row corresponds to the vector representation of a word in the vocabulary. During training, the weights are updated according to the gradient of the loss function.

Assuming the size of the vocabulary is $V$ and the embedding dimension is $D$, the embedding matrix $E = \{e_1, e_2, \ldots, e_V\}^T \in \mathbb{R}^{V \times D}$, where $e_i$ represents the embedding vector of the $i$-th word in the vocabulary. Each embedding vector has a dimension of $D$. For an input sequence of words $\{w_1, w_2, \ldots, w_T\}$, each word $w_t$ is mapped to an index $i_t$ in the vocabulary. The embedding layer retrieves the corresponding embedding vector from the weight matrix as:

$$x_t = E_{i_t} = e_{i_t} \tag{1}$$

where $x_t$ represents the embedding vector of the $t$-th word. After the completion of the model's forward propagation and the calculation of the loss function, the embedding matrix is updated by backpropagation using equation 2:

$$E_{i_t} \leftarrow E_{i_t} - \eta \frac{\partial L}{\partial E_{i_t}} \tag{2}$$

by combining equation 1 and equation 2, we can further express the update as:

$$E_{i_t} \leftarrow E_{i_t} - \eta \frac{\partial L}{\partial x_t} \tag{3}$$

The word embedding vectors contain rich semantic information about the respective words, and the cosine distance between vectors can capture the semantic divergence between words. Therefore, semantic segmentation problems can be addressed by computing the cosine similarity between the embedding vectors of each word using equation 4:

$$\text{Cosine Similarity} = \frac{A \cdot B}{\|A\| \|B\|} \tag{4}$$

$A \cdot B$ denotes the dot product of vectors $A$ and $B$, while $\|A\|$ and $\|B\|$ represent their Euclidean norms. The cosine similarity falls within the range *Cosine Similarity* $\in [-1, 1]$. A high cosine similarity between word embedding vectors indicates a strong semantic similarity or association between words, whereas a low cosine similarity suggests a significant semantic difference or lack of relevance. It is important to note that directly calculating the cosine similarity between every pair of words using equation 4 involves a computationally expensive operation. Assuming the embedding matrix $E \in \mathbb{R}^{n \times m}$, where $n$ is the number of words and $m$ is the embedding dimension of each word, the time complexity of calculating cosine similarity for all word pairs is $O(n^2 m)$. To mitigate this, we adopt a computational shortcut by sing equation 5 to compute a reference vector $R_f$, which is computed by averaging the embedding vectors of all the words in the sequence. This reduces the number of calculations required.

$$R_f = \frac{1}{n} \sum_{i=1}^{n} E_i \tag{5}$$

$R_f$ can be considered as the global semantic center of the entire text or corpus. We compute the cosine similarity between each word's embedding vector and the reference vector $R_f$. Through this approach, the time complexity is reduced to $O(nm)$, allowing us to efficiently assess the alignment of each word with the overall semantic context. This method not only lowers the computational complexity but also preserves the global semantic information.

### 3.2 CLUSTERING COMPRESSION AND CATEGORY CLASSIFICATION

With the reference vector $R_f$, we can quickly obtain a cosine similarity matrix $S \in \mathbb{R}^{b \times v}$, where $b$ represents the batch size and $v$ represents the sequence length. The core of the clustering operation is achieved through a masking mechanism. Based on predefined thresholds, the cosine similarity is segmented into intervals, and the corresponding mask matrix is generated from $S$. Through the weighted operation of the mask matrix, we can extract the embedding vectors that contain relevant semantic information.

For the cosine similarity, we assume the following: for an $n$-class classification task, the similarity can be divided into at most $n + 1$ segments. This means that for an $n$-class problem, the content can be segmented into $n + 1$ parts, corresponding to the content relevant to each of the $n$ classes and the content unrelated to these $n$ classes. In human learning, the process of classification often begins with detailed distinctions and gradually simplifies over time. Initially, due to insufficient

understanding of the classes, humans tend to divide the content into more detailed categories. However, with accumulated experience, the cognitive system evolves to adopt a more efficient strategy, reducing the number of classes and retaining only the most important distinctions (Žauhar et al., 2016; Constantinidis et al., 2023). We believe that under extreme conditions, complex tasks can only be simplified to binary classification decisions at most. This simplification mechanism aligns with Bayesian classification theory and the entropy minimization principle in information theory. It is important to note that by "extreme conditions," we mean that not all multidimensional classification problems can be fully reduced to binary classification. In tasks involving highly complex features, the simplification process may be constrained. And, (Wang et al., 2023a) demonstrated that for non-linear RNNs to approximate stable non-linear sequential relationships, the memory structure must exhibit exponential decay. Based on the above theories, we designed a cognitive function for FSFC to dynamically adjust the granularity of classification. The cognitive function is expressed as shown in equation 6:

$$C_n = C_0 - (C_0 - C_f) \times (1 - e^{-kn}) \tag{6}$$

Where $C_n$ represents the complexity at the $n$-th training epoch, $C_0$ is the initial complexity, and $C_f$ is the final complexity. $n$ denotes the current training epoch, and $k$ is the cognitive coefficient, which controls the rate at which cognitive complexity decreases. The introduction of this cognitive function enhances the model generalization ability.

Cosine similarity matrix not only helps the model perform effective semantic segmentation but can also be used to generate a mask matrix for clustering. To improve computational efficiency, we propose a method for generating the mask matrix by expanding the data dimensions and calculating the mask matrix in parallel. Specifically, based on predefined thresholds, the embedding vectors are divided into different similarity intervals, each corresponding to a mask matrix. All batches of mask matrices can be generated in a single computation. The generated mask matrix has the structure $(c, b, v, 1)$, where $c$ denotes the number of classes, $b$ denotes the batch size, and $v$ denotes the sequence length. Each mask matrix corresponds to a class and marks the words that belong to that class. By generating the mask matrices in batches, we can achieve clustering for multiple classes in a single operation.

Let $X \in \mathbb{R}^{b \times v \times d}$ represent the input embedding matrix of the text, and let $M_c \in \{0, 1\}^{b \times v}$ represent the mask matrix for class $c$, where $C$ is the total number of classes. Using equation 7, we obtain the compressed matrix $Z = \{Z_1, Z_2, \ldots, Z_c\} \in \mathbb{R}^{b \times C \times d}$:

$$Z_c = \sum_{i=1}^{v} X_i^{(c)} = \sum_{i=1}^{v} M_c^{(i)} \odot X^{(i)} \tag{7}$$

where $Z_c \in \mathbb{R}^{b \times d}$ is the weighted representation for class $c$, representing the weighted features for each batch. The compressed matrix $Z$ is then fed into the RNN for classification. Since the input text has been clustered and compressed, the sequence length of $Z$ is significantly reduced compared to the original input matrix, effectively alleviating the problem of gradient explosion or vanishing.

## 4 EXPERIMENTS

### 4.1 E-SCORE

To provide a comprehensive evaluation of the model's overall performance, we propose a new engineering evaluation metric called E-score. The E-score integrates both the model's accuracy and the time required for training.

Assuming there are $n$ models, with corresponding accuracy values $A = \{a_1, a_2, \ldots, a_n\}$ and training times $T = \{t_1, t_2, \ldots, t_n\}$, we first normalize the accuracy and time to eliminate the impact of differences in magnitude. The normalization of accuracy is given by equation 8:

$$\Delta_A = \frac{A}{\min(A)} \tag{8}$$

where $\Delta_A = \{\Delta_{a_1}, \Delta_{a_2}, \ldots, \Delta_{a_n}\}$ represents the relative improvement in accuracy of each model compared to the model with the lowest accuracy. This allows us to evaluate the model's performance

from the perspective of accuracy. For time efficiency, the normalization is conducted using equation 9:

$$\Delta_T = \frac{T}{\max(T)} \tag{9}$$

where $\Delta_T = \{\Delta_{t_1}, \Delta_{t_2}, \ldots, \Delta_{t_n}\}$ reflects the relative training efficiency of each model compared to the model with the longest training time. $\Delta_A$ and $\Delta_T$ indicate the time required for each model to achieve its respective improvement in accuracy. By taking $\Delta_T$ as the horizontal axis and $\Delta_A$ as the vertical axis, each model corresponding $(\Delta_t, \Delta_a)$ can be plotted on a two-dimensional coordinate plane.

The angle between the vector $(\Delta_t, \Delta_a)$ and the $x$-axis is denoted as $\theta = \{\theta_1, \theta_2, \ldots, \theta_n\}$, where $\theta \in \left(0, \frac{\pi}{2}\right)$. When the training time is constant, When the training time is constant, $\theta$ can be used to balance the trade-off between high and low accuracy. If the accuracy is the same, $\theta$ can balance the time efficiency of the models. However, there are certain limitations to the use of $\theta$. For example,when the vectors corresponding to two models are collinear, if the score is based solely on the value of $\theta$, a model with a short training time but lower accuracy could end up with the same score as a model with a long training time and higher accuracy, which is unreasonable. In practice, for any model, priority should always be given to accuracy. Only after ensuring that the minimum accuracy threshold is met should time efficiency be considered.

Therefore, the evaluation metric should primarily reflect the importance of accuracy. To achieve this, we transform the problem into polar coordinates, where $\theta$ is the polar angle and $\Delta_a$ is the radius. The E-score is then defined as the area of the sector formed by $\theta$ and the radius $\Delta_a$, as shown in equation 10:

$$E\text{-score} = \begin{cases} 0, & \text{if } A < A_{\text{threshold}} \\ \frac{1}{2}\theta\Delta_a^2, & \text{if } A \geq A_{\text{threshold}} \end{cases} \tag{10}$$

## 4.2 Test Performance on Multiple Datasets

In this section, we evaluate the performance of FSFC using several different classification datasets and compare it against LSTM and GRU. All models were implemented using the PyTorch framework. Specifically, the tests for the AG NEWS (Zhang et al., 2015), DBpedia (Auer et al., 2007), IMDB (Maas et al., 2011), and YahooAnswers (Zhang et al., 2015) datasets were conducted on an NVIDIA RTX 4090, while the tests for the YelpReviewFull (Zhang et al., 2015) and SogouNews (Zhang & LeCun, 2015) datasets were conducted on an NVIDIA RTX 3090. The maximum time step for all datasets was set to 400 to avoid any asynchronous effects on the experimental results.Table 1 presents the performance of three different models (LSTM, GRU, and FSFC) across various text classification datasets. The comparison primarily considers accuracy, training time, and the E-score metric. The results indicate that FSFC significantly improves training efficiency while maintaining competitive performance.

In terms of training time, FSFC demonstrates a significant advantage, consistently outperforming both LSTM and GRU with lower training times across all datasets. For instance, on the YelpReviewFull dataset, FSFC average training time per epoch is only 14.51 seconds, whereas LSTM requires 100.27 seconds, and GRU takes 125.97 seconds, representing efficiency improvements of 86% and 88%, respectively. On other datasets, such as YahooAnswers and IMDB, FSFC also shows a clear reduction in training time, making it particularly advantageous for large-scale text classification tasks.

With respect to accuracy, although FSFC shows slightly lower performance compared to LSTM and GRU, it outperforms LSTM on the YelpReviewFull dataset, achieving an accuracy of 50.34% compared to LSTM's 48.97%. On the AG NEWS and IMDB datasets, FSFC experiences a slight drop in accuracy but still maintains performance comparable to traditional methods. On the DBpedia dataset, FSFC's accuracy is same as both LSTM and GRU.

Regarding E-score metric, FSFC demonstrates remarkable performance in balancing training time and accuracy. For instance, on the AG NEWS dataset, FSFC achieves an E-score of 0.675, significantly higher than LSTM (0.456) and GRU (0.396). This indicates that while FSFC drastically reduces training time, it is still able to maintain accuracy comparable to or even exceeding traditional models. Overall, FSFC not only achieves accuracy comparable to LSTM and GRU but also signifi-

cantly reduces the total training time, making it better suited for scenarios with tight time constraints or limited computational resources.

Table 1: Comparison of accuracy and training time for FSFC, LSTM, and GRU across different datasets (batch size = 128, epochs = 100, learning rate = 0.001, time step = 400). Avg Time represents the average training time per epoch (in seconds), and Total Time represents the total training time for 100 epochs (in minutes).

| Dataset | Classes | Network | Accuracy | Avg Time | Total Time | E-score |
|---------|---------|---------|----------|----------|-----------|---------|
| AG NEWS | 4 | LSTM | 84.80% | 8.29 | 13.81 | 0.456 |
| | | GRU | 84.74% | 10.48 | 17.46 | 0.396 |
| | | FSFC | 84.40% | 2.34 | **3.91** | 0.675 |
| YahooAnswers | 10 | LSTM | 49.29% | 34.32 | 57.21 | 0.470 |
| | | GRU | 51.91% | 43.00 | 71.67 | 0.472 |
| | | FSFC | 48.40% | 5.96 | **9.93** | 0.717 |
| YelpReviewFull | 5 | LSTM | 48.97% | 100.27 | 167.11 | 0.449 |
| | | GRU | 52.11% | 125.97 | 209.95 | 0.462 |
| | | FSFC | 50.34% | 14.51 | **24.18** | 0.771 |
| DBpedia | 14 | LSTM | 66.21% | 13.58 | 22.64 | 0.437 |
| | | GRU | 66.21% | 16.25 | 27.08 | 0.393 |
| | | FSFC | 66.21% | 7.40 | **12.33** | 0.572 |
| IMDB | 2 | LSTM | 84.77% | 41.78 | 69.63 | 0.457 |
| | | GRU | 84.99% | 50.29 | 83.82 | 0.413 |
| | | FSFC | 83.40% | 12.98 | **21.50** | 0.660 |

## 4.3 PARAMETER SENSITIVITY

FSFC network introduces a cognitive coefficient $k$. To investigate the effect of different values of $k$ on the accuracy of the FSFC network, we conducted experiments across multiple datasets. Figure 2 shows the impact of different cognitive coefficient values $k$ on the test accuracy of the FSFC network on four datasets: AG NEWS, YahooAnswers, YelpReviewFull, and IMDB. The figure indicates that the sensitivity to $k$ varies across different datasets. For example, on the YahooAnswers dataset, accuracy significantly improves as the $k$ value increases, while on the YelpReviewFull dataset, smaller $k$ values yield better performance. By adjusting the $k$ value, the performance of the FSFC network can be further optimized.

## 4.4 EFFECTS OF TIME STEP

Time step is one of the key factors affecting the training efficiency of sequence models. As the time step increases, the sequence length the network needs to process grows, leading to higher computational costs. In this section, we focus on investigating the impact of different time steps on the training time of the FSFC network, and conduct a comparative analysis with traditional LSTM and GRU networks. We evaluated the training time trends of FSFC, LSTM, and GRU under different time steps (400, 500, 600, 700, and 800) using the SogouNews dataset. The SogouNews dataset, due to its longer text length and rich semantic information, provides a better platform for showcasing the performance differences of various models when handling long sequences. Moreover, the dataset contains a wide range of categories, which helps to assess the changes in training efficiency of each model when processing long-text sequences.

As shown in Figure 3, the training time for both LSTM and GRU networks increases significantly as the time step grows, while the FSFC network's training time remains nearly constant. The results clearly demonstrate that, compared to LSTM and GRU, FSFC is able to maintain a very low computational cost even when handling longer time sequences. As the time step increases, the training time for LSTM and GRU networks almost linearly increases, whereas FSFC's training time remains

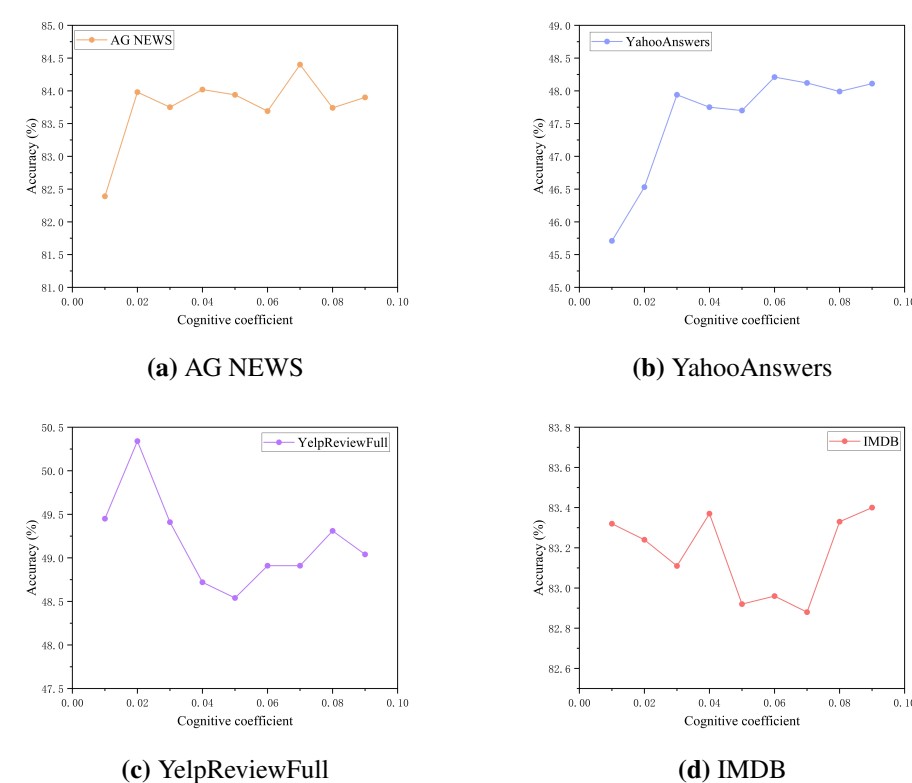

Figure 2: (a) The impact of the cognitive coefficient $k$ on the performance of the FSFC network for the AG NEWS dataset. (b) The impact of the cognitive coefficient $k$ on the performance of the FSFC network for the YahooAnswers dataset. (c) The impact of the cognitive coefficient $k$ on the performance of the FSFC network for the YelpReviewFull dataset. (d) The impact of the cognitive coefficient $k$ on the performance of the FSFC network for the IMDB dataset.

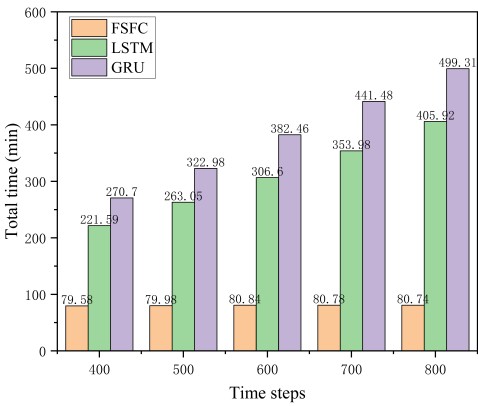

Figure 3: The time taken to train FSFC, LSTM, and GRU for 100 epochs under different time steps on the SogouNews dataset.

relatively stable. Therefore, using the FSFC network can significantly reduce training time, particularly when processing long-sequence text data. Its efficiency is especially prominent, effectively

overcoming the time bottleneck faced by traditional LSTM and GRU networks when handling long sequences.

## 5 CONCLUSION AND FUTURE WORK

In this study, we propose a novel recurrent neural network component, FSFC, designed specifically for text classification tasks, as a potential replacement for existing RNN components. FSFC effectively reduces sequence length by performing semantic clustering and compression on the text, which helps mitigate issues such as gradient vanishing or explosion. Unlike traditional recurrent neural networks, FSFC focuses on leveraging the short-term memory capability of RNNs. Furthermore, to comprehensively evaluate both the accuracy and training time of the network, we introduce a new evaluation metric, E-score, which combines model accuracy with training time, providing a more holistic measure of performance. Through the E-score, we are better able to assess the balance between accuracy and computational efficiency across different networks, particularly in scenarios where both model precision and time constraints must be considered.

FSFC network demonstrates slightly lower accuracy compared to LSTM and GRU, primarily due to its omission of positional information between words. In future work, we plan to design a positional encoder to incorporate word position information during the clustering and compression process, thereby improving the accuracy of the FSFC network.

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
