# OpenReview forum: "Fast Salient Factor Concentration (FSFC) Recurrent Neural Network for Text Classification"
_ICLR.cc/2025/Conference — ICLR 2025 Conference Withdrawn Submission_

### Official Review · Reviewer_rSDc · 2024-10-31

**Soundness:** 2
**Presentation:** 2
**Contribution:** 1
**Rating:** 3
**Confidence:** 4

**Summary:**

The paper proposes Fast Salient Factor Concentration (FSFC) RNN, a new architecture for classification tasks, to enhance the processing of crucial information by dynamically clustering and compressing semantic information. The performance on YelpReviewFull proves that FSFC has a higher accuracy with less training time.

**Strengths:**

1. The paper is well-organized, and the writing is fine.

2. The paper evaluates the performance of FSFC on four datasets. The experimental results show the effectiveness and efficiency of FSFC.

**Weaknesses:**

1. The motivation of this paper is not very clear. I'm not sure what problems this paper is trying to solve (Text classification? An alternative to RNN? An alternative to RNN on Text classification?).  It would be beneficial to specify what problems the authors aim to address—are they focusing on text classification, proposing an alternative to RNNs, or something else? Anyway, the contribution seems limited.

2. The idea presented in this paper does not appear particularly interesting. It proposes an alternative to RNNs for text classification, but I believe it lacks significant contributions to the current NLP community. Nowadays, many practitioners favor pre-trained models over RNNs for text classification tasks. Additionally, large language models (LLMs) tend to focus on multi-tasking rather than on single NLP tasks.

3. If the goal is to propose a new RNN, conducting experiments only on text classification is insufficient to verify the method's generalization.

4. Even for text classification tasks, the models compared in the paper are not comprehensive (e.g., ELMO, BERT, LLMs, and so on). The paper lacks comparisons with these strong methods for text classification. As I mentioned in Weakness 2, these models are precisely the kinds of models that are commonly utilized in the field of text classification today. If the paper focuses on text classification, not comparing with these mainstream models would be unfair.

**Questions:**

1. What is the motivation to propose an alternative to RNN for text classification? What problems are you trying to solve? Does "the long-term memory mechanisms of traditional RNNs do not fully align with human cognitive learning processes" really matter? How does it impact the performance in text classification tasks? Could you give us some concrete examples of how the misalignment between RNNs and human cognitive processes impacts performance in text classification tasks?

2. What is the contribution of FSFC? The author should discuss how FSFC compares to or complements more recent approaches in NLP. RNN-based methods for text classification are quite outdated now. While I am not against RNNs, achieving minor improvements and reducing training time seems to bring little new knowledge to the current NLP community.

3. How about employing other methods like BERT and LLMs? What is the value of an alternative to RNN compared with pretrained language models? Why do we still need RNN-based methods for text classification?

---

### Official Review · Reviewer_gMMw · 2024-11-04

**Soundness:** 1
**Presentation:** 1
**Contribution:** 2
**Rating:** 1
**Confidence:** 3

**Summary:**

This paper provides a method to remove the gating mechanisms of LSTM and condense memory.  The proposed method achieves better training speed while retaining accuracy.

**Strengths:**

This paper proposes a structure that is 5 times faster to train than LSTM.

**Weaknesses:**

1. It is tough to understand Figure 1. I can not find any RNN or recurrent in this figure. What is your whole network? Moreover, this figure doesn't have detailed captions.

2.  Why does long-term memory not matter in classification? Please provide justifications with citations or experiments.

3. This paper does not compare with other improvements on LSTM. Do other improvements over LSTM reduce the training time and achieve better accuracy?

4.  This paper does not provide details about the experiment setup.  Although we do not know what the network structure proposed in this paper looks like, nor do we know how the configuration compares to GRU and LSTM, or the length of the experimental data.

5. E-score is not more effective than just presenting accuracy and time.

**Questions:**

1. Does your model reduce the inference time?

---

### Official Review · Reviewer_MP7G · 2024-11-05

**Soundness:** 1
**Presentation:** 2
**Contribution:** 1
**Rating:** 3
**Confidence:** 4

**Summary:**

The paper introduces an RNN-based model called Fast Salient Factor Concentration (FSFC), designed to improve training efficiency in text classification by using short-term memory and semantic clustering. While FSFC's concept is intuitive, it has several critical limitations. The approach offers limited innovation, appearing incremental relative to existing RNN and attention-based techniques. Additionally, the research feels outdated, as NLP has largely moved toward transformer-based models. Performance evaluations yield mixed results: FSFC reduces training time but sacrifices accuracy on some datasets. Moreover, baseline comparisons are insufficient, with FSFC only benchmarked against LSTM and GRU models, excluding state-of-the-art transformers like RoBERTa and recent LLMs.

**Strengths:**

- Easy to follow and clearly written;
- Lightweight solution for text classification compared to transformer models.

**Weaknesses:**

- Limited novelty: The concept of text compression and segmentation is well-explored, with similar techniques in attention mechanisms and memory simplification.
- Outdated approach: It’s advisable for the authors to employ BERT-based models or LLMs, as transformers are now the NLP standard. Although addressing transformer complexity with RNNs is a good direction, applying it to basic tasks like text classification feels outdated.

**Questions:**

See weaknesses.

---

### Note · Authors · 2024-11-13

**Comment:**

Thank you to all the reviewers for their time and valuable feedback. The original intent of our method was to replace the basic LSTM or GRU modules in various complex models to accelerate training and inference speed. There are aspects of the paper that need improvement, and after further discussion, we have decided to withdraw the manuscript for further refinement. Once again, we sincerely appreciate the reviewers' efforts.

**Withdrawal Confirmation:**

I have read and agree with the venue's withdrawal policy on behalf of myself and my co-authors.